# Frequency and Voltage Supports by Battery-Fed Smart Inverters in Mixed-Inertia Microgrids

**Mohsen S. Pilehvar \*** and **Behrooz Mirafzal**

Department of Electrical and Computer Engineering, Kansas State University, Manhattan, KS 66506, USA; mirafzal@ksu.edu

**\*** Correspondence: pilehvar@ksu.edu

**Abstract:** This paper presents a piecewise linear-elliptic (PLE) droop control scheme to improve the dynamic behavior of islanded microgrids. Islanded microgrids are typically vulnerable to voltage and frequency fluctuations, particularly if a combination of high- and low-inertia power generation units are used in a microgrid. The intermittent nature of renewable energy sources can cause sudden power mismatches, and thus, voltage and frequency fluctuations. The proposed PLE droop control scheme can be employed in a battery energy storage system (BESS) to effectively mitigate voltage and frequency fluctuations in an islanded microgrid. Though the PLE shape can be implemented for any droop control scheme, it has been applied for active power-frequency (*P-f*) and reactive power-voltage (*Q-v*) droops in this paper. In addition, the dynamic response of a battery-fed smart inverter equipped with the proposed PLE droops has been compared with the results obtained from a linear droop control scheme in an islanded microgrid containing high- and low-inertia power-generation units. In this paper, the results of several case studies are presented to confirm the capability of the PLE droop control in mitigating voltage and frequency fluctuations in islanded microgrids.

**Keywords:** BESS; dynamic response; frequency fluctuations; voltage fluctuations; islanded mixed-inertia microgrids; PLE droop control

---

## 1. Introduction

Islanded microgrids, which typically fall in the category of weak grids [1], can suffer from voltage and frequency fluctuations caused by sudden changes in load demand, loss of power generation, and transitions between a grid-connected to an islanded mode of operation. The frequency oscillation is mainly due to the inherent low-inertia nature of inverter-based distributed generation (DG) units. The voltage oscillation can be due to the high impedance that inverters see from their terminals, circulating currents, and the interaction of phase-locked-loop (PLL) with current control loops [2,3]. Nevertheless, in grid-connected microgrids, since both voltage and frequency are tightly regulated by the main grid, the dynamic performance of the system is not as challenging as in islanded microgrids. Thus, the frequency and voltage stability in islanded microgrids is a technical challenge that needs to be addressed.

The generation side of islanded microgrids can be a mixture of synchronous-based and inverter-based DG units, forming islanded mixed-inertia microgrids. Diesel generators are one of the most common synchronous-based DGs which have been extensively utilized in islanded microgrids, particularly in rural areas [4–6]. The use of diesel generators is because of their low cost and relatively high reliability [7,8]. In the presence of diesel generators, inverter-based DGs fed by renewable energy resources, e.g., photovoltaic (PV) panels and BESSs, can be employed to reduce the fuel consumption [9–12]. In such mixed-inertia microgrids, the diesel generator typically operates in grid-forming mode by taking responsibility for controlling the voltage and frequency using an automatic

voltage regulator (AVR) and governor, when other DGs can work in *PQ*-controlled or grid-following mode. The grid-following inverters can either operate in grid-feeding mode or grid-supporting mode when also offering ancillary services [13–16]. If the diesel generator is out of service for any reason, battery-fed inverters can operate as the grid-forming units.

Many studies have been reported on the voltage and/or frequency support in islanded microgrids. In [17–19], an enhanced frequency response is achieved by coordinating the operation of DG units. However, coordinating different DG units requires a centralized algorithm with additional communication links. In [20], a piecewise droop curve for inverter-based DG units in islanded microgrids is presented, in which the droop coefficients are adjusted based on an optimal power flow with the aim of minimizing the power losses, while satisfying the power balance, frequency and current constraints. References [21,22] utilize BESS for frequency regulation and stability improvement of islanded microgrids by optimizing the size of BESS and employing virtual inertia, respectively. Furthermore, comprehensive analyses on the impact of the droop characteristics of BESS on the frequency response of the system are presented in [23,24]. On the other hand, different approaches are presented in [25,26] to use PV units instead of BESS for decreasing the system cost and frequency variations at the same time. However, the impacts of control strategies developed in [25,26] are limited to the availability of solar power. Moreover, an approach is presented in [27] which reduces the dependency of islanded microgrids on BESS by controlling the frequency through voltage regulation.

Nevertheless, the aforementioned methods in [17–27] focus only on the frequency regulation of the system. However, maintaining the AC-bus voltage within acceptable limits is essential for improving the stability, reliability, and power quality of the system, especially in the presence of critical loads [28,29]. In [30], an enhanced voltage profile is achieved by synchronizing the response times of different voltage regulation devices employed in an islanded microgrid. The network theory concepts are utilized in [31] to develop a control scheme for mitigating the voltage fluctuations of an AC-bus in islanded microgrids. In [32,33], the voltage and frequency of the system are controlled by BESS instead of a diesel generator. However, due to the low inertia of BESS, this might lead to the stability deterioration of the system. In order to enhance the voltage and frequency profiles during abnormalities, an approach is presented in [34] which coordinates the DG units of an islanded microgrid. However, according to the provided results, the proposed method is not applicable during the transition from islanded to grid-connected mode.

This paper presents an approach to enhance the dynamic behavior of islanded microgrids by suppressing the voltage and frequency fluctuations. To attain this, BESS is used to inject and absorb the active and reactive power during power transients. A PLE *P-f* droop is suggested for BESS, which enhances the frequency profile of the system. Similarly, in order to improve the voltage profile of the system, a PLE *Q-v* droop is introduced. Furthermore, it is demonstrated that for any linear droop characteristic with a specified droop coefficient, there is also a PLE characteristic which can be formulated using the proposed method in this paper. Compared with conventional linear droops, the proposed PLE droop results in injecting and absorbing more active and reactive power by BESS, which compensates for the instantaneous power mismatches between the generation and consumption during transients.

The proposed PLE droop can quickly react to frequency and voltage fluctuations in comparison with the conventional nonlinear and linear *P-f* and *Q-v* droop controls. A nonlinear droop results in injecting and absorbing less amount of active and reactive power during transients. This means that the system will need more time to restore the voltage and frequency after severe transients. The nonlinear droop method is typically utilized in DC and AC microgrids for achieving an enhanced power sharing between different DGs during the steady-state condition [35–37]. The proposed PLE side by side to the conventional nonlinear and linear *P-f* droops are shown in Figure 1. In this figure, the frequency in an islanded microgrid, $f_M$, is regulated at its desired value, $f_M^*$, while a similar set of curves can be imagined for the main AC-bus voltage in the microgrid, $v_M$. The linear trajectories can also be implemented with a dead band around the desired voltage and frequency, see Figure 1d,

in which the system performance is very dependent on the length of the dead band, and larger dead bands might cause abrupt changes in the active and reactive power of BESS while the voltage and frequency of the AC-bus are going inside and outside of the dead band. In contrast, the presence of the narrow linear regions in the proposed PLE droops helps the BESS-based DG smoothly operate in a transient condition.

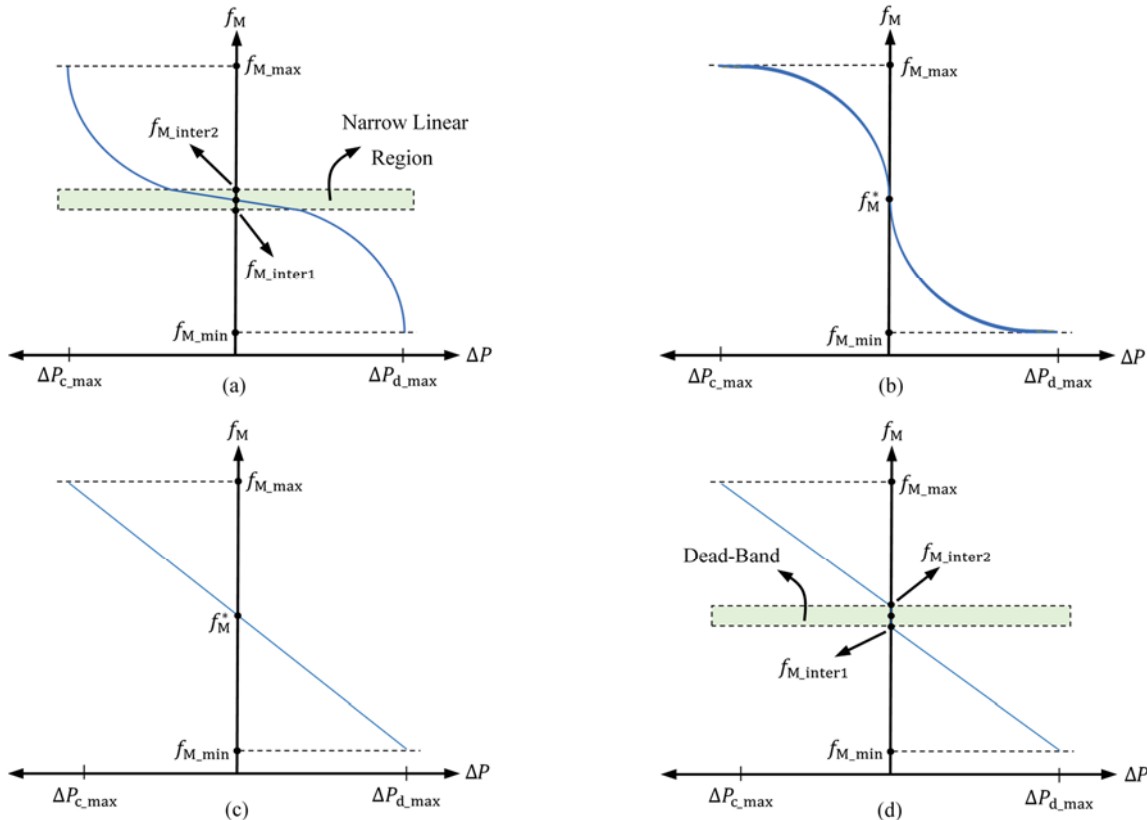

**Figure 1.** Piecewise linear-elliptic (PLE) active power-frequency (*P-f*) characteristic versus other *P-f* characteristics for the dynamic-response enhancement in islanded microgrids: (**a**) PLE; (**b**) nonlinear; (**c**) linear; and (**d**) linear with a dead band.

The rest of this paper is organized as follows. Section 2 introduces the islanded mixed-inertia microgrid under study equipped with both linear and PLE droops. The frequency and voltage dynamics of the system during transients are studied in Sections 3 and 4, respectively, along with the detailed explanation of the proposed PLE droops for an increase in the mitigation of frequency and voltage variations. Different case studies are provided in Section 5 to compare the efficacy of linear and PLE droops in a dynamic-response enhancement of islanded microgrids. Finally, Section 6 concludes this paper.

## 2. System Description

The schematic of an islanded mixed-inertia microgrid under study is depicted in Figure 2. Due to the investigation of the cooperative role of BESS-based DG in improving the dynamics of the system, the controller of BESS is represented with more detail. The primary voltage and frequency regulation of the AC-bus is the main responsibility of a diesel generator using a governor and AVR, i.e., in grid-forming mode. In order to provide the desired active and reactive power for the system, the PV unit is operating in *PQ*-control mode or grid-following mode. However, due to the considerable difference between the inertia of a diesel generator and a PV unit, the power mismatch between the

supply and demand can easily happen. This leads to voltage and frequency deviations which might have severe consequences on critical loads.

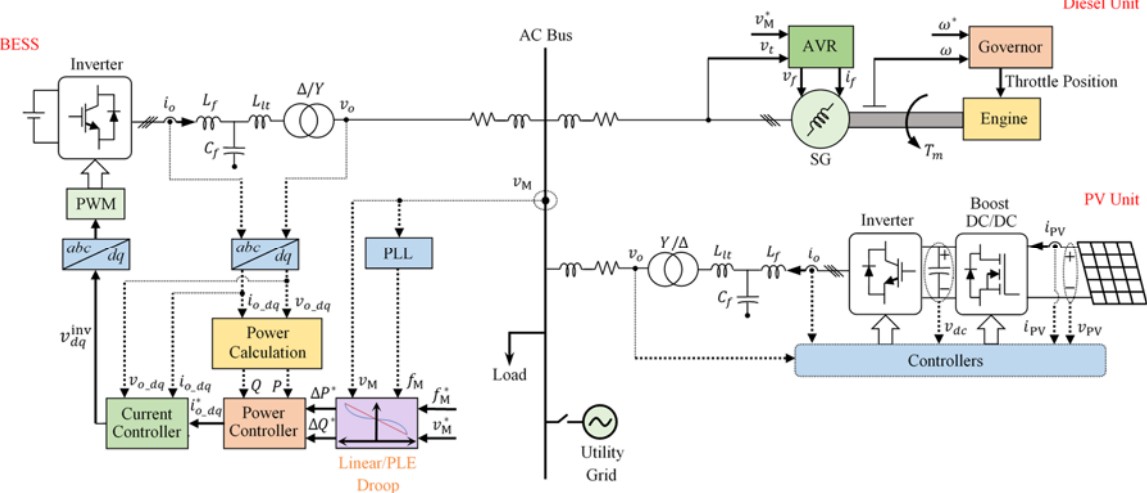

**Figure 2.** Schematic of the islanded mixed-inertia microgrid under study.

In order to resolve the short-term voltage and frequency variation issues, a BESS-based DG is employed to balance the power generation and consumption by injecting and absorbing the active and reactive power. To this end, two different PLE droop characteristics, i.e., *P-f* and *Q-v*, are defined for BESS to cooperate in voltage and frequency regulation in the case of abnormalities. During the steady-state condition, the voltage and frequency of the AC-bus, i.e., $v_M$ and $f_M$, are regulated at their desired values, i.e., $v_M^*$ and $f_M^*$, and thus, based on the implemented droop characteristics, the BESS-based DG is not designed to cooperate in the steady-state voltage and frequency regulation. However, since $v_M^*$ and $f_M^*$ are not fixed at their desired values during transients, this DG can participate in voltage and frequency restoration. Under such conditions, this DG injects and absorbs the active and reactive power based on the proposed approach, resulting in an improved dynamic response compared with linear characteristics, as shown in Section 5.

## 3. Frequency Dynamic-Response Improvement

This section investigates the solutions to achieve an enhanced frequency response for islanded microgrids during transients. First, a brief overview of the frequency dynamic-response of the system along with implemented *P-f* droops, i.e., linear and PLE, is presented, and then a further explanation of the proposed PLE characteristic is provided.

### 3.1. Frequency Dynamic-Response of the System

The frequency variations of the AC-bus during the settling time of $t_s$ are depicted in Figure 3a. These frequency oscillations can be a result of abnormal conditions such as the step change in load demand, variations of solar irradiance, and transitions from grid-connected to islanded modes. At point A, the system operates at its desired frequency, $f_M^*$. Nevertheless, during the transients, the AC-bus frequency oscillates. As an example, at point B, the frequency of an AC-bus is lower than its desired value, while at point C, the system suffers from high frequency. Consequently, as shown in Figure 3b, in order to reduce these short-term variations, linear and PLE *P-f* characteristics can be employed in BESS to compensate for the active power imbalance between supply and demand. In Figure 3b, the trajectory of a linear *P-f* droop is depicted by the red line, and the blue line is the trajectory of the introduced PLE droop. Both linear and PLE droops are defined within the same frequency and active power range. In other words, they both have the same start point and end point. Comparing linear and PLE trajectories leads to the fact that employing the PLE droop results in injecting/absorbing more

active power during abnormalities, for the faster adjustment of the frequency variations. This leads to an improved frequency dynamic response and stability enhancement of the overall system.

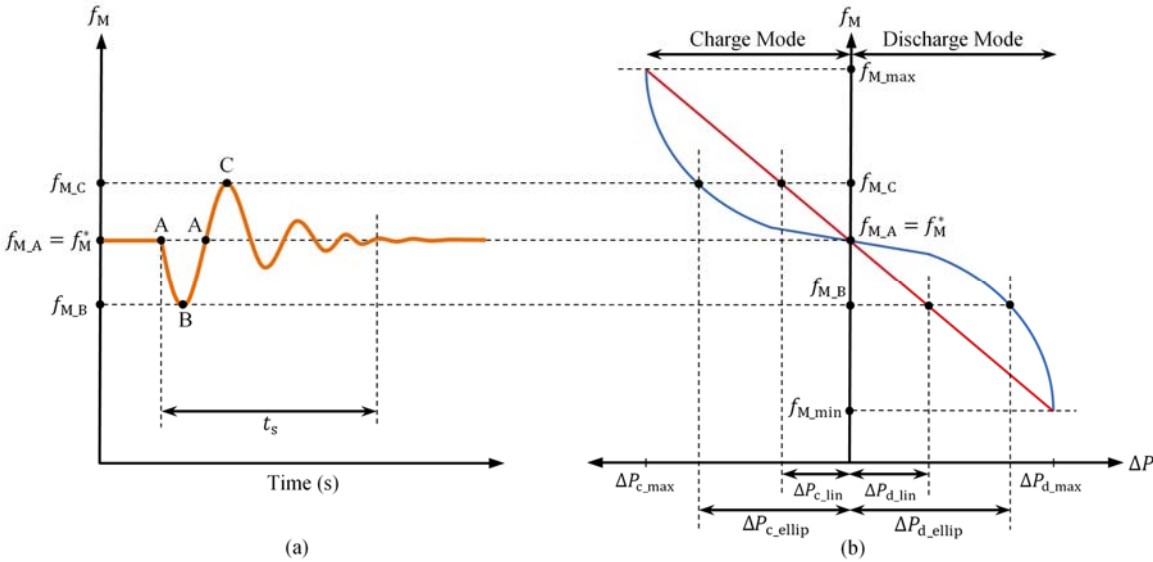

**Figure 3.** (**a**) Frequency deviations of the AC-bus during abnormal conditions, and (**b**) corresponding linear (red line) and PLE (blue line) *P-f* characteristics.

Depending on the frequency of the AC-bus, three operation modes can be considered for BESS: discharging mode, charging mode, and idle mode. During under frequency, i.e., $f_M < f_M^*$, the BESS-based DG operates in discharging mode, i.e., $\Delta P > 0$, to provide the deficit active power caused by considerable inertia difference between DGs. Moreover, during over frequency, i.e., $f_M > f_M^*$, the BESS-based DG operates in charging-mode, i.e., $\Delta P < 0$, and absorbs the extra active power in the system to maintain the frequency. During the steady-state condition, the system works at its desired frequency, i.e., $f_M = f_M^*$, and hence, the BESS-based DG is in idle mode, i.e., $\Delta P = 0$.

According to Figure 3b, the injected and absorbed active power by BESS for the linear *P-f* characteristic with the droop coefficient of $m_P$ can be obtained as follows:

$$\Delta P_{\text{lin}} = \frac{f_M^* - f_M}{m_P} \tag{1}$$

where $f_M$ and $f_M^*$ are the frequency of AC-bus and its desired value, respectively. The injected power, $\Delta P_{\text{lin}}$, must be positive (battery discharging mode) for $f_M < f_M^*$, i.e., $\Delta P_{\text{lin}} = \Delta P_{\text{d\_lin}} > 0$, and the absorbed power must be negative (battery charging-mode) for $f_M > f_M^*$, i.e., $\Delta P_{\text{lin}} = \Delta P_{\text{c\_lin}} < 0$.

### 3.2. PLE P-f Droop

The upper and lower limits of frequency, i.e., $f_{M\_max}$ and $f_{M\_min}$, and the range of active power for BESS-based DG are predefined. Therefore, a possible solution to improve the frequency dynamic-behavior of the system is to establish a new trajectory instead of a linear trajectory. In discharging mode, the start point, i.e., $f_M^*$, and end point, i.e., $f_{M\_min}$, can be connected by a quarter ellipse centered at $f_{M\_min}$, whose horizontal radius is $\Delta P_{d\_max}$ and the vertical radius is $f_M^* - f_{M\_min}$. Herein, $\Delta P_{d\_max}$ is the maximum active power that can be provided by BESS, which can be calculated from (1) as follows:

$$\Delta P_{\text{d\_max}} = \frac{f_M^* - f_{M\_min}}{m_P} > 0 \tag{2}$$

Therefore, considering the aforementioned quarter ellipse, the active power provided by BESS during the discharge mode can be obtained as follows:

$$\Delta P_{d\_ellip} = \sqrt{\frac{\left(f_M^* - f_{M\_min}\right)^2 - \left(f_M - f_{M\_min}\right)^2}{m_P^2}} > 0 \tag{3}$$

Similarly, the trajectory between $f_M^*$ and $f_{M\_max}$ during the charging can be defined as a quarter ellipse centered at $f_{M\_max}$, with the horizontal radius of $\Delta P_{c\_max}$ and vertical radius of $f_{M\_max} - f_M^*$. Considering the specified value of droop coefficient, i.e., $m_P$, $\Delta P_{c\_max}$ is the maximum active power that can be absorbed by BESS, and can be obtained from (1) as follows:

$$\Delta P_{c\_max} = \frac{f_M^* - f_{M\_max}}{m_P} < 0 \tag{4}$$

As a result, the active power absorbed by BESS during the charge mode can be formulated as follows:

$$\Delta P_{c\_ellip} = -\sqrt{\frac{\left(f_M^* - f_{M\_max}\right)^2 - \left(f_M - f_{M\_max}\right)^2}{m_P^2}} < 0 \tag{5}$$

Since the slope of elliptic trajectory is zero at the desired frequency, a narrow linear region should be considered around the desired frequency as follows:

$$\Delta P_{lin2} = \frac{f_M^* - f_M}{m_{P2}} \tag{6}$$

where $m_{P2}$ is the droop coefficient of a narrow linear trajectory that should constantly be less than $m_P$.

## 4. Voltage Dynamic-Response Improvement

This section presents the PLE droop to achieve an enhanced voltage profile for islanded microgrids during transients. The frequency and voltage dynamic supports are added to the smart inverter fed by BESS as ancillary services, see Figure 4. The PLE droop for voltage is very similar to PLE for the frequency dynamic supports, but for the continuity in the discussion, PLE for voltage is briefly formulated in the following.

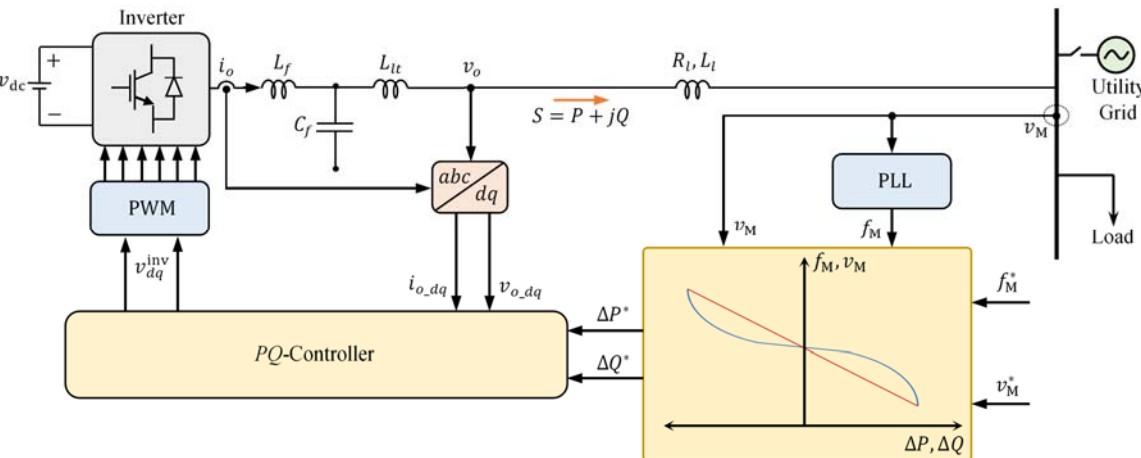

**Figure 4.** Control block diagram of a *PQ*-controlled BESS-based inverter equipped with PLE and linear droops.

The continuity in the voltage variations of the AC-bus leads to significant operational challenges for ensuring the stability of the system. Figure 5 shows these voltage fluctuations during severe conditions, and the corresponding linear, i.e., the red line, and PLE, i.e., the blue line, $Q$-$v$ droops. Similar to the previous section, both linear and PLE droops have the same start point and end point. Besides, utilizing the PLE droop results in injecting/absorbing more reactive power during transients, leading to an improved voltage profile of the system. At point A, the system operates at its desired voltage, i.e., $v_M = v_M^*$, and thus, the BESS is in idle mode, i.e., $\Delta Q = 0$. However, at point B, the AC-bus voltage falls below its desired value, i.e., $v_M < v_M^*$, and hence, the BESS-based DG operates in capacitive mode, i.e., $\Delta Q > 0$, to inject the deficit reactive power into the system. On the other hand, at point C, the AC-bus voltage increases over its desired value, i.e., $v_M > v_M^*$, and the BESS-based DG operates in inductive mode, i.e., $\Delta Q < 0$, to restore voltage by absorbing the extra reactive power in the system.

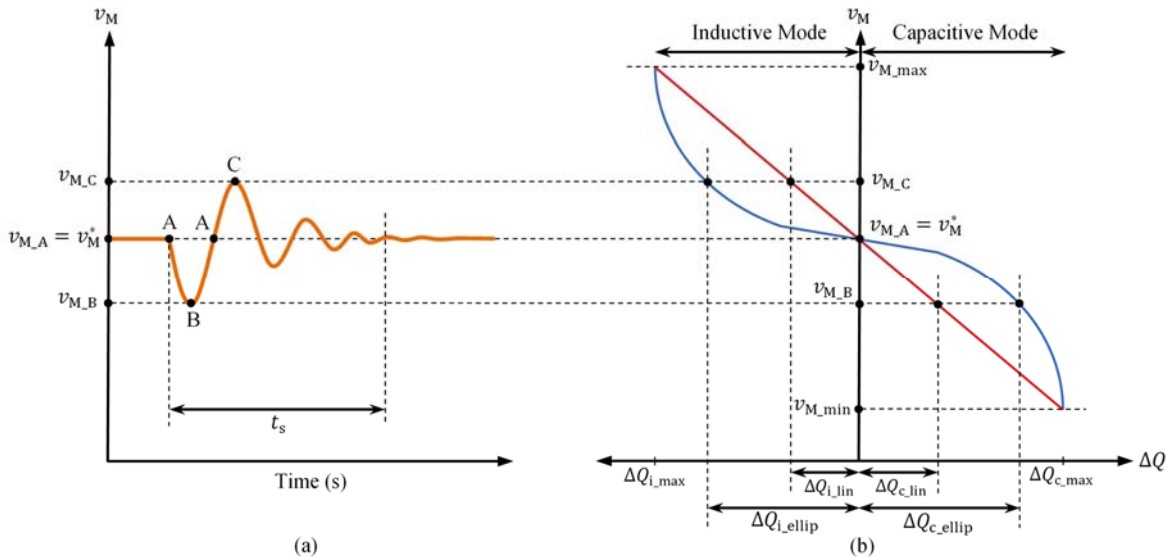

**Figure 5.** (**a**) Voltage deviations of the AC-bus during abnormal conditions, and (**b**) the corresponding linear (red line) and PLE (blue line) *P-f* characteristics.

For the linear $Q$-$v$ droop with the droop coefficient of $m_Q$, i.e., the red line in Figure 5b, the injected/absorbed reactive power by BESS can be calculated as follows:

$$\Delta Q_{\mathrm{lin}} = \frac{v_M^* - v_M}{m_Q} \tag{7}$$

where $v_M$ and $v_M^*$ are the voltage of the AC-bus and its desired value, respectively. The injected reactive power, $\Delta Q_{\mathrm{lin}}$, must be positive (battery capacitive-mode) for $v_M < v_M^*$, i.e., $\Delta Q_{\mathrm{lin}} = \Delta Q_{\mathrm{c\_lin}} > 0$, and the absorbed reactive power must be negative (battery inductive-mode) for $v_M > v_M^*$, i.e., $\Delta Q_{\mathrm{lin}} = \Delta Q_{\mathrm{i\_lin}} < 0$. Similarly, the PLE $Q$-$v$ characteristic can be formulated as follows:

$$\Delta Q_{\mathrm{c\_ellip}} = \sqrt{\frac{\left(v_M^* - v_{M\_min}\right)^2 - \left(v_M - v_{M\_min}\right)^2}{m_Q^2}} > 0 \tag{8}$$

$$\Delta Q_{\mathrm{i\_ellip}} = -\sqrt{\frac{\left(v_M^* - v_{M\_max}\right)^2 - \left(v_M - v_{M\_max}\right)^2}{m_Q^2}} < 0 \tag{9}$$

Notice, the slope of the elliptic trajectory is zero at the desired voltage. Thus, a narrow linear region is defined around the desired voltage as follows:

$$\Delta Q_{\text{lin2}} = \frac{v_{\text{M}}^{*} - v_{\text{M}}}{m_{Q2}} \tag{10}$$

where $m_{Q2}$ is the droop coefficient of a narrow linear trajectory which should always be chosen less than $m_{Q}$.

## 5. Results and Discussion

In the following subsections, different case studies are presented to compare the performance of linear and PLE droops. The detailed model of islanded microgrid, shown in Figure 2, has been developed in an PSCAD/EMTDC environment. The droop coefficients are given in Table 1. Due to the fast response of BESS, the linear (with and without a dead band) and PLE characteristics are implemented in it to inject the active/reactive power during the under frequency/voltage, and absorb the active/reactive power in case of the over frequency/voltage.

**Table 1.** System parameters.

| | Parameter | Value |
|---|---|---|
| Distributed Generation (DG) Units | Battery operating voltage range | 360–600 V |
| | Battery rated energy | 101 kWh |
| | PV maximum power | 50 kW |
| | Diesel generator maximum power | 127 kW |
| | Diesel generator inertia constant ($H$) | 0.3 s |
| Linear and PLE Droops | $f_{\text{M}}^{*}$ | 60 Hz |
| | $f_{\text{M\_min}}$ | 58 Hz |
| | $f_{\text{M\_max}}$ | 62 Hz |
| | $v_{\text{M}}^{*}$ (line-to-line) | 480 V |
| | $v_{\text{M\_min}}$ (line-to-line) | 400 V |
| | $v_{\text{M\_max}}$ (line-to-line) | 560 V |
| | $m_{P}$ | 0.05 Hz/kW |
| | $m_{P2}$ | 0.02 Hz/kW |
| | $m_{Q}$ | 13 V/kVAr |
| | $m_{Q2}$ | 4 V/kVAr |

### 5.1. Linear Droop with and without Dead Band

The first case study investigates the dynamic performance of the system with and without the implementation of dead band in linear droop characteristics. Due to the islanding operation of the microgrid, the primary regulation of voltage and frequency is the main responsibility of a diesel generator. The active and reactive power demand, i.e., $P_{\text{D}}$ and $Q_{\text{D}}$, are initially 30 kW and 10 kVAr. At $t = 150$ s, the active and reactive power demand step up to 120 kW and 30 kVAr. As one can observe in Figure 6, both voltage and frequency profiles are improved in the absence of dead bands, which is due to the injection/absorption of more active and reactive power during transients. It can be observed that the frequency overshoot decreases from 60.27 to 60.05 Hz, while the voltage drop is increased from 384.8 to 410.4 V.

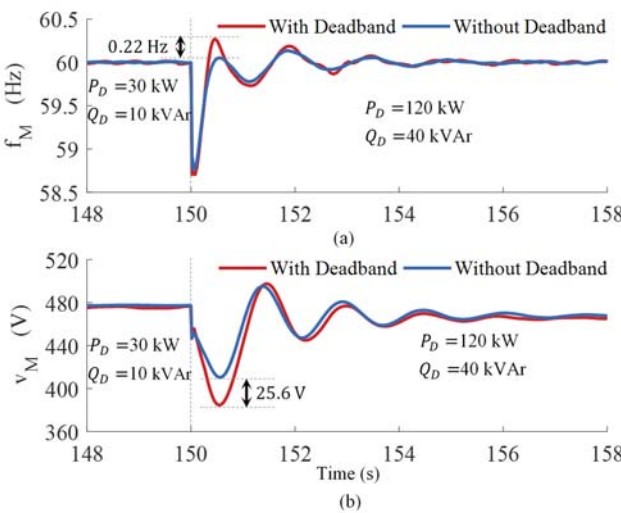

**Figure 6.** (**a**) Frequency and (**b**) voltage variations of the AC-bus in the presence and absence of dead bands in linear droops when the load demand is stepped up.

*5.2. PLE Droop versus Linear Droop for Load Step Change*

This case study compares the effect of employing linear and PLE droops on the dynamic performance enhancement of the system when there is a step change in load demand. The load step change is the same as the previous case study, and as shown in Figure 7a,b, the proposed PLE droops help the system to have enhanced frequency and voltage responses with shorter settling times. Compared with the case of a linear *P-f* droop, implementing the PLE *P-f* droop results in improving the frequency nadir by about 0.25 Hz, see Figure 7a. Moreover, as illustrated in Figure 7b, the fluctuations of the AC-bus voltage are significantly mitigated by employing the proposed PLE *Q-v* droop, while the voltage amplitude is reduced by about 20 V. This enhancement in the dynamic response of the system is because of the injecting/absorbing more active and reactive power by BESS, i.e., $\Delta P$ and $\Delta Q$, when the PLE droops are applied, see Figure 7c,d. The active and reactive power of the load, i.e., the $P_L$ and $Q_L$, are shown in Figure 7e,f, respectively, comparing the linear and PLE droops. It can be observed that by utilizing the PLE droop, the system needs less amount of time to meet the new load demand, while the active and reactive power oscillations of the load are mitigated. Table 2 summarizes the comparison between the effect of utilizing linear and PLE droops on the dynamic performance enhancement of islanded microgrid, based on the results demonstrated in Figure 7. From Table 2, utilizing the proposed PLE droops results in a remarkable improvement in the dynamic behavior of the microgrid for a step change in load demand.

**Table 2.** Dynamic performance comparison of linear and PLE droops for a step change in load demand.

|  | With Linear | | With PLE | |
|---|---|---|---|---|
|  | **Settling Time** | **Over Shoot** | **Settling Time** | **Over Shoot** |
| $f_M$ | 3 s | 0.1371 Hz | 2 s | 0.0472 Hz |
| $v_M$ | 4 s | 30 V | 2.5 s | 10 V |
| $P_L$ | 4 s | 17 kW | 3 s | 9 kW |
| $Q_L$ | 4 s | 5 kVAr | 3 s | 2 kVAr |

*5.3. PLE Droop versus Linear Droop for the Transition from Grid-Connected to Islanded Mode*

The vulnerability of islanded microgrids to the voltage and frequency variations compared with grid-connected microgrids is due to the presence of low-inertia power generation units. The viability of proposed PLE droops in reducing the short-term voltage and frequency fluctuations during the

transition from grid-connected to islanded mode is investigated in this case study. The load demand does not change during this case study ($P_D = 60$ kW and $Q_D = 20$ kVAr). The microgrid is initially connected to the grid, and at $t = 150$ s, the grid operator disconnects it from the main grid.

Figure 8 compares the effect of employing linear and PLE droops in the dynamic-response enhancement of a microgrid. As can be found from Figure 8a, with the implementation of a PLE *P-f* droop, the frequency is restored faster, while the frequency overshoot is decreased from 61.1 Hz to 60.8 Hz. Similarly, as shown in Figure 8b, the proposed PLE *Q-v* droop suppresses the voltage fluctuations of an AC-bus during the transition from grid-connected to islanded mode, while the voltage amplitude is reduced from 540 V to 523 V. Moreover, without the implementation of PLE droop, the system requires almost 9 s to restore the voltage. However, the proposed PLE droop helps the system to restore the AC-bus voltage in less than 5 s, see Figure 8b. As illustrated in Figure 8c,d, when the system is connected to the grid, the injected/absorbed active and reactive power by BESS, i.e., $\Delta P$ and $\Delta Q$, are almost equal to zero, and this is due to the tight regulation of both voltage and frequency at their desired values by the utility grid. Nevertheless, after transferring to islanded mode, the BESS starts injecting/absorbing active and reactive power. It can be seen that more active and reactive power are injected/absorbed by BESS when the PLE droops are employed, leading to a significant reduction in voltage and frequency variations. Finally, the fluctuations of $P_L$ and $Q_L$ are analyzed in Figure 8e,f, respectively, comparing the linear and PLE droops. It can be seen that equipping BESS with the proposed PLE droops leads to a significant reduction in fluctuations of $P_L$ and $Q_L$ during transients. Table 3 investigates the effect of utilizing the proposed PLE droops in the dynamic performance improvement of a microgrid based on the results demonstrated in Figure 8. The provided comparison analysis in Table 3 indicates that the proposed PLE droops are an effective solution for dealing with the stability challenges and improving the dynamic behavior of islanded microgrids during the transition from grid-connected to an islanded mode of operation.

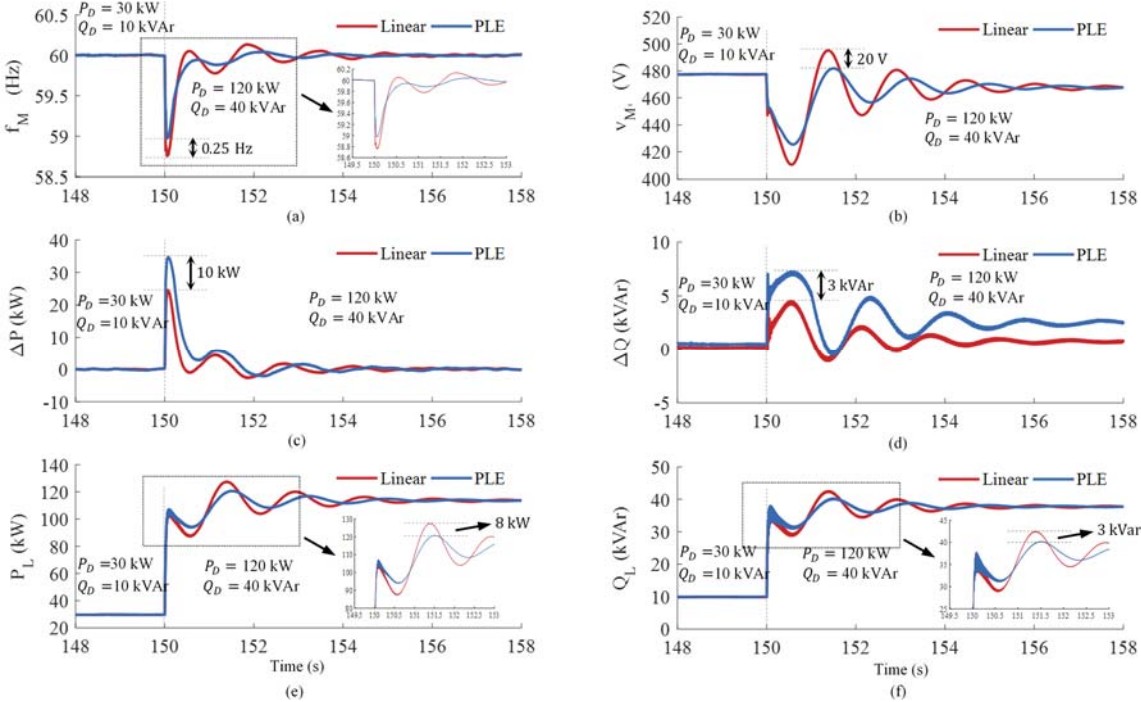

**Figure 7.** Comparing the performance of linear and PLE droops in a dynamic-response enhancement of the system during a step change in load demand: (**a**) AC-bus frequency; (**b**) AC-bus voltage; (**c**) BESS active power; (**d**) BESS reactive power; (**e**) load active power; and (**f**) load reactive power.

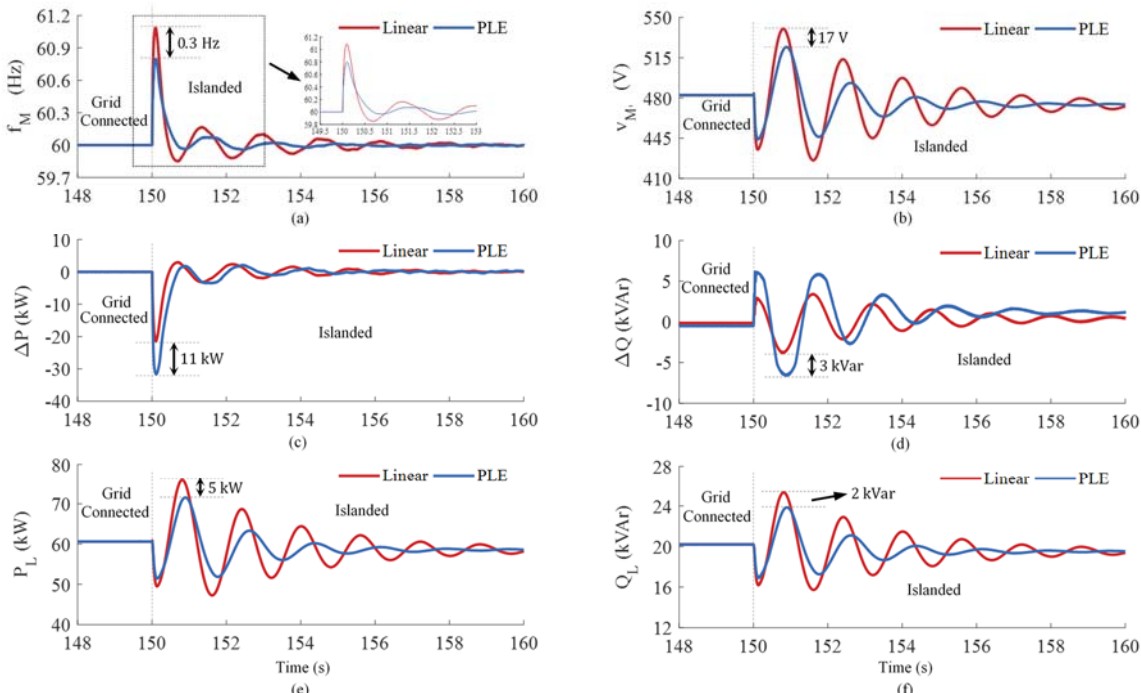

**Figure 8.** Comparing the performance of linear and PLE droops in the dynamic-response enhancement of the system during the transition from grid-connected to islanded mode: (**a**) AC-bus frequency; (**b**) AC-bus voltage; (**c**) BESS active power; (**d**) BESS reactive power; (**e**) load active power and (**f**) load reactive power.

**Table 3.** Dynamic performance comparison of linear and PLE droops during transition from grid-connected to islanded mode.

|  | With Linear | | With PLE | |
|---|---|---|---|---|
|  | Settling Time | Over Shoot | Settling Time | Over Shoot |
| $f_M$ | 7 s | 1.1 Hz | 3 s | 0.8 Hz |
| $v_M$ | 9 s | 70 V | 5 s | 53 V |
| $P_L$ | 8 s | 15 kW | 4 s | 10 kW |
| $Q_L$ | 9 s | 6 kVAr | 4 s | 4 kVAr |

## 5.4. Change in the Length of Narrow Linear Regions

This case study investigates the effect of change in the length of narrow linear regions on the dynamic behavior of the system. As shown in Figures 1 and 3–5, the proposed PLE droops include narrow linear regions around the desired voltage and frequency. The presence of narrow linear regions prevents the abrupt changes in the active and reactive power injected/absorbed by BESS for small variations of voltage and frequency. As the result, a smoother operation by BESS during short-term transients can be achieved. Figures 9 and 10 demonstrate the effect of change in the droop coefficients of narrow linear regions, i.e., $m_{P2}$ and $m_{Q2}$, on the mitigation of the frequency and voltage variations, respectively, when there is a step change in load demand. An increase in $m_{P2}$ and $m_{Q2}$ results in an increase in the length of narrow linear regions. In other words, increasing $m_{P2}$ and $m_{Q2}$ adds more linearity to the PLE droops which results in the injection/absorption of a lesser amount of active and reactive power during transients. Thus, in order to achieve an enhanced dynamic behavior, smaller values of $m_{P2}$ and $m_{Q2}$ are preferable. However, due to the sudden changes in the active and reactive power of BESS for very small values of $m_{P2}$ and $m_{Q2}$ (or extremely narrowed linear regions), the minimum allowable values of $m_{P2}$ and $m_{Q2}$, i.e., $m_{P2\_min}$ and $m_{Q2\_min}$, are required. Figures 9

and 10 illustrate $m_{P2\_min}$ and $m_{Q2\_min}$, at which there is a trade-off between the linearity of the PLE droops and the dynamic-response enhancement of the system.

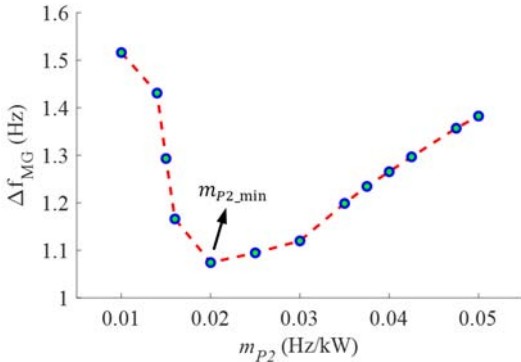

**Figure 9.** Effect of the change in $m_{P2}$ (or the length of a narrow linear region) on the frequency variations of an AC-bus when $m_P = 0.05$ Hz/kW.

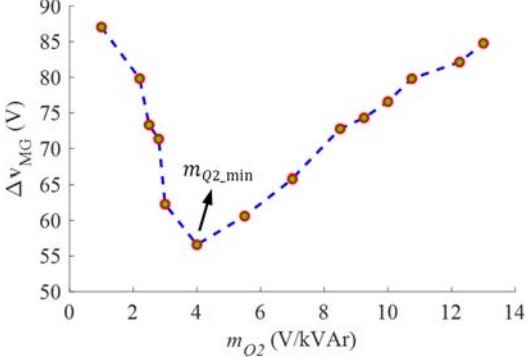

**Figure 10.** Effect of the change in $m_{Q2}$ (or the length of a narrow linear region) on the voltage variations of an AC-bus when $m_Q = 13$ V/kVAr.

## 6. Conclusions

In this paper, the supportive role of BESS in enhancing the dynamic behavior of islanded microgrids during abnormalities was investigated. Due to the high implementation of low-inertia DG units, islanded microgrids are vulnerable to voltage and frequency variations. In addition, due to the presence of both synchronous-based and inverter-based DGs with a considerable difference in inertia, a power mismatch between supply and demand during transients is inevitable. This leads to voltage and frequency deviations with severe consequences such as blackouts. To this end, two PLE droops, i.e., *P-f* and *Q-v*, have been suggested for BESS to suppress the voltage and frequency variations during abnormalities such as a load step change and transition from grid-connected to islanded mode. Utilizing the proposed droops turns the conventional inverter employed in BESS into a smart inverter capable of coping with short-term transients. It has been indicated that any linear droop characteristic with a specified droop coefficient can be replaced by a PLE droop using the proposed approach in this paper. In comparison with conventional linear droops, equipping the BESS with proposed PLE droops has resulted in a considerable reduction in voltage and frequency deviations. Several case studies have been carried out to confirm the merits of presented PLE droops in the dynamic-performance enhancement of islanded microgrids.

**Author Contributions:** Conceptualization, M.S.P.; writing—original draft, M.S.P.; writing—review and editing, B.M.; validation, M.S.P.; supervision, B.M. All authors have read and agreed to the published version of the manuscript.

**Funding:** This research received no external funding.

**Conflicts of Interest:** The authors declare no conflict of interest.

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
