# Peer review of "Frequency and Voltage Supports by Battery-Fed Smart Inverters in Mixed-Inertia Microgrids"

_electronics, doi:10.3390/electronics9111755_

Round 1

Reviewer 1 Report

The paper presents a piecewise linear-elliptic (PLE) droop control scheme to enhance the dynamic behavior of islanded microgrids.

In general, the paper is well written and the conclusions are supported by the results of the simulations.

This reviewer has only two concerns:

  • how your proposal can match with the grid codes prescribing linear drop?
  • why your proposal is applied only to islanded microgrids?

Some minor amendments:

- "dc" and "ac" must be "DC" and "AC" throughout the manuscript.

- Regarding non-linear droop, for completing the review of the state of the art, you should discuss also: "Driven Primary Regulation for Minimum Power
Losses Operation in Islanded Microgrids", DOI: 10.3390/en11112890.

Reviewer 2 Report

The manuscript deals with a topic that worth to be further investigated. It can be accepted for publication but the authors must conduct changes in order to improve its quality.

The size and quality of figures must be improved. It is very hard to read the majority of them.

The Introduction must be extended. The authors must present the general research area to unfamiliar readers and at most to present the current state-of-the-art in order to show the contribution/novelty of their work. Authors must describe/analyse more the current mentioned references and to include more recent references, such as the following:

Vita V., Alimardan T., Ekonomou L., The impact of distributed generation in the distribution networks’ voltage profile and energy losses, Proceedings of the 9th IEEE European Modelling Symposium on Mathematical Modelling and Computer Simulation, Madrid, Spain, pp. 260-265, 2015.

Eliassi M., Torkzadeh R., Mazidi P., Rodriguez P., Pastor R., Vita V., Zafiropoulos E., Dikeakos C., Michael M., Tapakis R., Boultadakis G., Conflict of interests between SPC-based BESS and UFLS scheme frequency responses, 21st International Symposium on High Voltage Engineering (ISH 2019), Budapest, Hungary, (DOI) 10.1007/978-3-030-37818-9_6, 2019.

Torkzadeh R., Eliassi M., Mazidi P., Rodriguez P., Brnobić D., Krommydas K.F., Stratigakos A.C., Dikeakos C., Michael M., Tapakis R., Vita V., Zafiropoulos E., Pastor R., Boultadakis G., Synchrophasor based monitoring system for grid interactive energy storage system control, 21st International Symposium on High Voltage Engineering (ISH 2019), Budapest, Hungary, (DOI) 10.1007/978-3-030-37818-9_9, 2019.

A discussion section that will comment on the produced results must be included.

Conclusions must summarize the work presented within the paper.

Round 2

Reviewer 2 Report

The authors have conducted the requested changes.

The paper has been significantly improved.

It can be accepted for publication.